



# An urban trees parameterization for modelling microclimatic variables and thermal comfort conditions at street level with the Town Energy Balance model (TEB-SURFEX v8.0)

Redon Emilie[1], Lemonsu Aude[1], and Masson Valéry[1]

[1]CNRM, Météo-France/CNRS, 42 avenue Gaspard Coriolis, 31057 Toulouse cedex, France

**Correspondence:** Aude Lemonsu (aude.lemonsu@meteo.fr)

**Abstract.** The TEB urban climate model has recently been improved to more realistically address the radiative effects of trees within the urban canopy. These processes necessarily have an impact on the energy balance that needs to be taken into account. This is why a new method for calculating the turbulent fluxes for sensible and latent heat has been implemented. This method remains consistent with the "bigleaf" approach of the ISBA model which deals with energy exchanges between vegetation and atmosphere within TEB. Nonetheless, the turbulent fluxes can now be dissociated between ground-based natural covers and tree stratum above (knowing the vertical leaf density profile), which can modify the vertical profile in air temperature and humidity in the urban canopy. In addition, the aeraulic effect of trees is added, parameterized as a drag term and an energy dissipation term in the evolution equations of momentum and of turbulent kinetic energy, respectively. This set of modifications relating to the explicit representation of tree stratum in TEB is evaluated on an experimental case study. The model results are compared to micrometeorological and surface temperature measurements collected in a semi-open courtyard with trees and bordered by buildings. The new parameterizations improve the modelling of surface temperatures of walls and pavements thanks to taking into account radiation absorption by trees, and of air temperature. The wind speed is strongly slowed down by trees that is also much more realistic. The universal thermal climate index diagnosed in TEB from inside-canyon environmental variables is highly dependent and sensitive to these variations in wind speed and radiation. This demonstrates the importance of properly modelling interactions between buildings and trees in urban environments, especially for climate-sensitive design issues.

## 1  Introduction

The urban climate commonly refers to the modification of local climate by the urban environment. It results from the establishment of radiative, energetic, dynamic, hydrological surface processes that are pecular to urban covers properties (Oke et al., 2017). This urban climate may however present important spatial variabilities within the city. The street-level meteorological variables, i.e. air temperature, humidity, wind, are modified in by the local environment depending the morphology and arrangement of buildings, the surface properties and more generally the land covers composition (Houet and Pigeon, 2011;





Fenner et al., 2014; Ndetto and Matzarakis, 2014; Alexander and Mills, 2014).

The presence of vegetation, and especially urban trees, is one element of modification of local microclimate (Bernatzky, 1982; Oke, 1989; Shashua-Bar and Hoffman, 2000; Potchter et al., 2006; Shashua-Bar et al., 2011). The trees functioning is governed by own physiological processes together with local environmental conditions. When located in a street canyon, the trees come in interaction with the surrounding urban elements. The radiative exchanges are modified as well as energy fluxes and local air flows. The incoming radiation is intercepted by the tree foliage depending on its localization and coverage in the street (Souch, 1993; Armson et al., 2012; Berry et al., 2013). It is for one part reflected upward according to reflective properties of leaves, whereas the other part is partially transmitted through the foliage or absorbed by it. The multiple reflections of short- and long-wave radiation between all components of the canyon are also disrupted by the presence of trees that are furthermore an additional source of infrared emissions. A part of energy absorbed by trees is used in the transpiration process associated to photosynthesis (Peters et al., 2011; Qiu et al., 2013). This is a water exchange from aerial parts of plants to the atmosphere through stomata. The transpiration rate is related to the stomata opening level which is regulated according to environmental conditions of sunlight, temperature, humidity and wind, but also to soil water availability (Konarska et al., 2016; Litvak and Pataki, 2016). As a consequence, the energy exchanges that are mostly dominated by heat transfers through conduction and convection in built-up environments can be significantly modified with the vegetation effects (Grimmond et al., 1996; Offerle et al., 2006; Best and Grimmond, 2016). Finally, the trees are physical obstacles to the flow within the street (Heisler, 1990; Giometto et al., 2017; Martini et al., 2017). They lead to a drag effect on the mean flow and modify local turbulent exchanges which can have an impact on ventilation as well as on particles dispersion (Buccolieri et al., 2011; Abhijith et al., 2017).

Some of urban climate models are capable to simulate the presence of certain vegetated elements in the urban environment. They are not represented with the same level of accuracy, nor the same physical processes. The high spatial resolution softwares or models that are based on near-reality numerical mock-ups of the environment integrate trees as fully elements composing the scene simulated as porous media (Bruse and Fleer, 1998; Salim et al., 2015). They are obstacles on the same level as buildings, and can consequently be involved in dynamical processes in CFD modelling. They are resolved explicitly in some radiative transfer models such as SOLENE (Miguet and Groleau, 2007) or DART (Gastellu-Etchegorry et al., 1996). They are here described as turbid objects that intercept, transmit, absorb, and emit radiation, in complex interaction between all surrounding objects of the scene.

Some of urban canopy models coupled or implemented in meso-scale atmospheric models now include a representation of urban vegetation and even of trees. This is done through a simplified approach for which trees effects are parameterized (Lee and Park, 2008; Krayenhoff et al., 2014, 2015; Ryu et al., 2016; Redon et al., 2017). Despite a simplified description of the urban environment, the main physical processes can be taken into account, i.e. radiation interactions between a mean tree foliage layer within the canyon and surrounding urban facets, the modification of wind profile within the canyon by drag effect of this foliage layer, or the transpiration of trees. These models offer the interest to be able to apply over the whole city and to operate in complete three-dimensional atmospheric simulations where two-ways interactions between complex surfaces and atmosphere are resolved. They have then the capability to simulate a certain level of microclimate variability between neigh-





bourhoods (de Munck et al., 2018) especially according to the presence of vegetation and trees, and the potential influences or interferences between local microclimates by horizontal advection.

## 2 Representation of natural covers in TEB

### 2.1 Previous developments and general approach

The Town Energy Balance (TEB) urban canopy model is one of the first model from this generation to have included urban vegetation in local-scale interaction with the built-up elements. It has been progressively made more complex by integrating new types of nature elements and new associated processes. The first step was the inclusion of ground-based vegetation within the canyon by implementing the Interaction Soil-Biosphere-Atmosphere (ISBA, Noilhan and Planton, 1989) model within TEB (Lemonsu et al., 2012). ISBA is a surface-vegetation-atmosphere transfer (SVAT) model. Through this coupling, it makes

possible to simulate the physiological behaviour of plants subject to radiative effects of urban geometry and to microclimate conditions of the urban environment. Reversely, the microclimate within the canyon can be impacted by the evapotranspiration from vegetation at the ground. A module of extensive green roofs was also developed (de Munck et al., 2013) still using the ISBA model to simulate the hydrological and energetic functioning of the green roofs, as well as the energy exchanges with atmosphere and the thermal coupling with the buildings on which they are installed.

### 15 2.2 Radiative effects of urban trees

Finally the most recent developments regard trees. Redon et al. (2017) proposed a new parameterization for modelling the radiative effects of trees by including a tree-foliage stratum within the urban canyon which can partially cover ground-based surfaces. The positioning and geometry of individual trees are not described explicitly but rather approached as (1) an horizontal coverage fraction in within the canyon, (2) a vertical thickness by defining a mean height of trees and a mean height

of trunks, and (3) a mean leaf density profile (see description in Figure 1). The trees foliage can intercept a part of incoming radiation depending on canyon geometry, that is either reflected upward, or transmitted through the foliage, or absorbed by the foliage. In addition, the foliage layer takes part of radiation inter-reflections within the canyon between the different components (trees, walls, road, ground-based natural covers) and contributes to infrared emissions. Separated calculations are done for direct and diffuse components of shortwave radiation, and for long-wave radiation. The direct shortwave radiation

is assumed directional whereas the diffuse shortwave radiation, the longwave radiation, and any radiation after reflection are assumed to be isotropic. The radiation interactions between all components of the canyon are consequently managed using form factors calculated between all of them.

These developments were evaluated by comparison with solar enlightenment modelling performed by a high-resolution archi-

tectural software, for a large set of simple-geometry urban canyons, with various aspect ratios and various trees arrangements within (Redon et al., 2017). A general defect is the underestimation of fluxes intercepted by the tree-foliage stratum that is





explained by the fact that TEB does not represent the sides of crown. The simplified approach is also a limitation for describing some vegetation arrangements such as multiple lanes of trees. Nonetheless, the results are quite acceptable and confirm this new version make possible to better simulate the radiative interactions in canyons with trees.

In coherence with this explicit separation of vegetation strata, there was then a need to adapt the calculation of the turbulent fluxes related to low and high vegetation as well as to include the drag effect of trees on wind profile in the canyon. These new developments and their evaluation by comparison to microclimatic measurements are presented and discussed here. There are complemented by an updated calculation of the universal thermal comfort index (UTCI) by including the effect of trees.

## 3    Surface energy balance of the canyon components in TEB

### 3.1    Descripion of natural covers with the "bigleaf approach"

The TEB model resolves the radiative budget for each component of the urban canyon. In the case of a treeless canyon, it accounts for obstruction effects due to buildings in calculation of incoming short- and long-wave radiation interception by roads, walls, and natural ground-based surfaces, as described in details in Masson (2000) and Lemonsu et al. (2012). The new version of Redon et al. (2017) now includes the additional interactions with the tree-foliage stratum. From the resolution of

radiation budget, the energy quantity absorbed in short- ($S^*$) and long-wave ($L^*$) radiation by each component or net radiation ($Q^*$) is determined:

$$Q^* = S^* + L^* \tag{1}$$

This net energy source is redistributed as turbulent sensible ($Q_H$) and latent ($Q_E$) heat fluxes between each considered component and local atmosphere, and as a storage heat flux by conduction ($Q_G$) through the component itself (i.e. through the

artificial-materials layers that compose the roof, the road or the walls, or in the ground for the natural covers).

For the built-up facets of urban canyon (road, wall, roof), the surface energy balance (SEB) calculations remain unchanged in the TEB model, but with $Q^*$ potentially modified in case of canyons with trees. Since the works of Lemonsu et al. (2012), the turbulent processes for natural parts of the urban canyon (natural soils and ground-based vegetation) are resolved in TEB

through the integration of the ISBA model that is here constrained by radiative and microclimatic conditions related to the urban environment. The ISBA model is based on that is called "the bigleaf approach" in which the natural covers are managed as a unique composite compartment. This compartment consists in fractions of bare soil, low vegetation, and high vegetation (and possibly snow) according to which mean radiative, aerodynamic, and physiologic parameters are defined. A single temperature ($T_{nat}$) is associated to it, and a single SEB is resolved:

$$Q^*_{nat} = Q_{H_{nat}} + Q_{E_{nat}} + Q_{G_{nat}} \tag{2}$$





This net radiation depends on the radiation budget which is expressed as following:

$$Q^*_{nat} = S^*_{nat} + L^*_{nat} = (1 - \alpha_{nat})S^{\downarrow}_{nat} + \epsilon_{nat}(L^{\downarrow}_{nat} - \sigma T^4_{nat}) \tag{3}$$

The incoming short- and long-wave radiation intercepted by natural covers compartment ($S^{\downarrow}_{nat}$ and $L^{\downarrow}_{nat}$) are calculated for a reference level at the ground within the urban canyon. The composite albedo ($\alpha_{nat}$) is calculated as an average of bare soil and vegetation albedo (which are themselves average albedos of snow-covered and snow-free surfaces). Same is done for the emissivity ($\epsilon_{nat}$).

### 3.2 Modification of surface energy balance due to implementation of trees

The implementation of a tree canopy in TEB like a supplementary foliage stratum makes possible to separate the incoming radiation received by natural covers as a part received by ground-based natural surfaces ($S^{\downarrow}_g$ and $L^{\downarrow}_g$) and a part received by trees ($S^{\downarrow}_t$ and $L^{\downarrow}_t$), and also to compute the net radiation for both of them ($S^*_t$ and $L^*_t$). For purpose of simplicity, the bigleaf concept used in ISBA for the SEB calculation and the resolution of the surface-layer temperature evolution equation is here remained. This requires to calculate aggregated radiation fluxes intercepted by natural covers which is provided to ISBA. Considering the foliage stratum partially overlaps the ground-based surfaces, these fluxes are aggregated according to a single fraction of natural covers, and weighted to insure energy conservation as following:

$$S^{\downarrow}_{nat} = \frac{\delta_g S^{\downarrow}_g + \delta_t S^{\downarrow}_t}{\delta_g + \delta_t} \tag{4}$$

$$L^{\downarrow}_{nat} = \frac{\delta_g L^{\downarrow}_g + \delta_t L^{\downarrow}_t}{\delta_g + \delta_t} \tag{5}$$

Here, $\delta_g$ is the ground-based surface fraction of the canyon covered by gardens (i.e. bare soil and low vegetation) and $\delta_t$ is the overlapping fraction of tree stratum. Note that these fractions are not dependant one from the other, so that their sum can be greater than 1 (Figure 1).

Both $Q_{H_{nat}}$ and $Q_{E_{nat}}$ fluxes calculated by ISBA for the composite compartment are then desaggregated in two contributions from ground-based natural covers ($Q_{H_g}$ and $Q_{E_g}$) and from trees ($Q_{H_t}$ and $Q_{E_t}$), based on ponderation coefficients related to the cover fractions:

$$Q_{H_g} = \left(\frac{\delta_g}{\delta_g + \delta_t}\right) Q_{H_{nat}} \tag{6}$$

$$Q_{H_t} = \left(\frac{\delta_t}{\delta_g + \delta_t}\right) Q_{H_{nat}} \tag{7}$$

$$\tag{8}$$

Same equations are applied for $Q_{E_t}$ and $Q_{E_t}$.



## 4 Inclusion of trees in the surface boundary layer parameterization of TEB

### 4.1 Principle of the surface boundary layer parameterization

The TEB-SBL (SBL referred to as surface boundary layer) parameterization has been implemented in TEB in order to improve the meteorological variable prediction within the urban canyon (Hamdi and Masson, 2008; Masson and Seity, 2009; Lemonsu
et al., 2012). TEB-SBL resolves the surface boundary layer for an air volume in the canyon from a system of evolution equations for air temperature ($T$), specific humidity ($q$), wind speed ($U$), and turbulent kinetic energy ($E$). For taking into account the effects of canyon on the local atmospheric characteristics evolution, an additional forcing term is included in each of these equations, according to the approach proposed by Yamada (1982) for the drag forces related to the vegetation canopies. The equations have the same general expression, with $V$ the considered variable and $F_V$ the general forcing term including
advection, Coriolis force, and pressure gradient:

$$\frac{\partial V}{\partial t} = F_V + \left.\frac{\partial V}{\partial t}\right|_{can} \tag{9}$$

The last term to the right is the forcing term due to the canyon. It translates a drag force for the wind, a heating/cooling effect for air temperature, a humidification/dryness effect for air humidity, and a dissipation/production effect for turbulent kinetic energy. According to Martilli et al. (2002), these additional contributions are associated to horizontal and vertical surfaces of
the canyon:

$$\left.\frac{\partial V}{\partial t}\right|_{can} = \left.\frac{\partial V}{\partial t}\right|_{can}^{H} + \left.\frac{\partial V}{\partial t}\right|_{can}^{V} \tag{10}$$

For these terms, the equations are resolved according to a vertical discretisation of the air volume from the ground to a reference atmospheric level located above the top of buildings (Figure 1). As described by Lemonsu et al. (2012), the equation system of the TEB-SBL parameterization is expressed as following for each $k$ vertical layer :

$$\left.\frac{\partial U(k)}{\partial t}\right|_{can} = -C_{d_{bld}} U(k)^2 \frac{S_{V_w}(k)}{V_{air}} - u_*^2 \left( \frac{S_{H_R}(k)}{V_{air}} + \frac{S_{H_r}(k)}{V_{air}} \right) \tag{11}$$

$$\left.\frac{\partial E(k)}{\partial t}\right|_{can} = C_{d_{bld}} U(k)^3 \frac{S_{V_w}(k)}{V_{air}} \tag{12}$$

$$\left.\frac{\partial T(k)}{\partial t}\right|_{can} = \frac{Q_{H_R}}{\rho C_p} \cdot \frac{S_{H_R}(k)}{V_{air}} + \frac{Q_{H_r}}{\rho C_p} \cdot \frac{S_{H_r}(k)}{V_{air}} + \frac{Q_{H_{nat}}}{\rho C_p} \cdot \frac{S_{H_{nat}}(k)}{V_{air}} + \frac{Q_{H_w}}{\rho C_p} \cdot \frac{S_{V_w}(k)}{V_{air}} \tag{13}$$

$$\left.\frac{\partial q(k)}{\partial t}\right|_{can} = \frac{Q_{E_R}}{\rho \mathcal{L}_v} \cdot \frac{S_{H_R}(k)}{V_{air}} + \frac{Q_{E_r}}{\rho \mathcal{L}_v} \cdot \frac{S_{H_r}(k)}{V_{air}} + \frac{Q_{E_{nat}}}{\rho \mathcal{L}_v} \cdot \frac{S_{H_{nat}}(k)}{V_{air}} \tag{14}$$

where $C_{d_{bld}}$ is the drag coefficient for buildings, $u_*$ the friction velocity, $V_{air}$ the air volume of the SBL-scheme layer where
exchanges take place, $\rho$ is the air density and $\mathcal{L}_v$ is the latent heat for vaporization. The sensible heat fluxes of vertical surfaces ($Q_{H_w}$ for walls) contribute to air temperature evolution at layer $k$ relatively to the fraction of the total wall surface in contact with the considered air layer ($S_{V_w}(k)$). The sensible heat fluxes of horizontal surfaces combine $Q_{H_R}$, $Q_{H_r}$, and $Q_{H_{nat}}$ for roofs, road, and natural covers, respectively. The roofs only contribute to air temperature for vertical level above building top (with surface $S_{H_R}$). The contributions of road and of natural covers are here included at the ground-level so that they affect





only the first layer of TEB-SBL (with respective surfaces $S_{H_r}(k)$ and $S_{H_{nat}}(k)$). The same types of contributions are parameterized for humidity but in the form of latent heat fluxes.

The implementation of trees in TEB requires to modify the ensemble of equations in order to take into account the verti-
cal redistribution of turbulent fluxes and the drag effect of the foliage layer.

## 4.2   Distribution of heat and humidity fluxes from natural covers

The turbulent fluxes of trees ($Q_{H_t}$ and $Q_{E_t}$) coming from the desaggregation of ISBA fluxes calculated with the bigleaf concept
(see Eq 8) are assumed to be produced not at ground-level but at height of the foliage layer within the canyon. These fluxes
have therefore to be vertically distributed in the TEB-SBL equations that is here parameterized according to a function of the
foliage density contained in each $k$ layer:

$$Q_{H_t}(k) = \int_k LAD(z)dz \ Q_{H_t} = \left(\frac{d_t(k)}{h_t - h_{tk}}\right) LAI \ Q_{H_t} \tag{15}$$

$$Q_{E_t}(k) = \int_k LAD(z)dz \ Q_{E_t} = \left(\frac{d_t(k)}{h_t - h_{tk}}\right) LAI \ Q_{E_t}$$

with $d_t(k)$ is the foliage-layer thickness in the $k$ layer. The leaf area index ($LAI$ in m$^2$ m$^{-2}$) is prescribed as input data. The
leaf area density ($LAD$ in m$^2$ m$^{-3}$) is the vertical profile of $LAI$ which depends on the total thickness of the foliage layer
and the form of tree crowns. For now, tree crowns are described as cylinders so that $LAD$ is constant over the thickness of
tree-foliage layer i.e. between $h_{tk}$ and $h_t$ that are the height of trunks and of trees, respectively.

These turbulent fluxes are now included in the evolution equations of temperature and humidity profiles of TEB-SBL, in
addition to ground-based contributions coming from natural soils and low-level vegetation:

$$\left.\frac{\partial T(k)}{\partial t}\right|_{can} = \frac{Q_{H_R}}{\rho C_p}\frac{S_{H_R}(k)}{V_{air}} + \frac{Q_{H_r}}{\rho C_p}\frac{S_{H_r}(k)}{V_{air}} + \frac{Q_{H_g}}{\rho C_p}\frac{S_{H_g}(k)}{V_{air}} + \frac{Q_{H_w}}{\rho C_p}\frac{S_{V_w}(k)}{V_{air}} + \frac{Q_{H_t}(k)}{\rho C_p}\frac{V_{grid}}{V_{air}}\delta_t \tag{16}$$

$$\left.\frac{\partial q(k)}{\partial t}\right|_{can} = \frac{Q_{E_R}}{\rho \mathcal{L}_v}\frac{S_{H_R}(k)}{V_{air}} + \frac{Q_{E_r}}{\rho \mathcal{L}_v}\frac{S_{H_r}(k)}{V_{air}} + \frac{Q_{E_g}}{\rho \mathcal{L}_v}\frac{S_{H_g}(k)}{V_{air}} + \frac{Q_{E_t}(k)}{\rho \mathcal{L}_v}\frac{V_{grid}}{V_{air}}\delta_t \tag{17}$$

with $\delta_t$ the overlapping fraction of trees in the canyon, and $V_{grid}$ the total air volume.

## 4.3   Aerodynamic effect of trees

The presence of trees also modifies the air flow within the canyon. For account to this, a supplementary drag term is now
included in the evolution equations of momentum and turbulent kinetic energy:

$$\left.\frac{\partial U}{\partial t}\right|_{can} = -C_{d_{bld}}U(k)^2\frac{S_{V_w}(k)}{V_{air}} - u_*^2\frac{S_{H_R}(k)}{V_{air}} - C_{d_t}U(k)^2 LAD(k)\delta_t \tag{18}$$

$$\left.\frac{\partial E}{\partial t}\right|_{can} = C_{d_{bld}}U(k)^3\frac{S_{V_w}(k)}{V_{air}} + C_{d_t}U(k)^3 LAD(k)\delta_t \tag{19}$$





Numerous studies found in literature (Cassiani et al., 2008; Dupont and Brunet, 2008; Aumond et al., 2013; Krayenhoff et al., 2015) propose an optimized value of the drag coefficient of trees ($C_{d_t}$). Until the works of Katul et al. (1998), this coefficient is usually defined as a constant $C_{d_t} = 0.20$.

## 5 Parameterization of universal thermal climate index

5 ### 5.1 General principle

The UTCI calculation according to the polynomial regression equation proposed by Bröde et al. (2012) (not detailed here) requires four meteorological parameters: air temperature at 2 m above the ground, the water vapor pressure at the same level, the wind speed at 10 m above the ground, and the mean radiant temperature ($T_{mrt}$). This equation has been implemented in TEB (Kwok et al., 2019, supplementary materials) for calculating three UTCIs that are associated with a person (1) in the street 10 exposed to the sun, (2) in the street in the shadow, and (3) in the building.

For outdoor conditions in case of a person in the sun, $T_{mrt}$ is calculated by accounting for the ensemble of radiation sources received by the person, i.e. the direct and diffuse incoming short-wave radiation ($S^\downarrow$ and $S^\Downarrow$), the short-wave radiation after reflection on the walls, the road, and the ground-based natural covers in the canyon ($S_w^r$, $S_r^r$, $S_g^r$), the incoming atmospheric 15 long-wave radiation ($L^\downarrow$), and the infrared emissions from surrounding canyon surfaces ($L_w^\uparrow$, $L_r^\uparrow$, $L_g^\uparrow$ for walls, roads, and ground-based natural covers).

The direct short-wave radiation assumed to be unidirectional is weighted by a factor of projected area relative to the person ($f_p$) which depends on sun elevation ($\gamma$ in ˚) according to the formulation of Fanger (1970):

$$f_p = 0.308\, cos\left\{\gamma\left(1 - \frac{\gamma^2}{48402}\right)\right\} \tag{20}$$

The other fluxes that are considered to be isotropic in TEB radiative calculations are weighted by the form factors calculated for the person ($b$ index is used for $body$) in relation to the surrounding elements that contribute to radiation, i.e. $\Psi_{bs}$, $\Psi_{br}$, $\Psi_{bw}$ for the sky, the road, and the walls, respectively (ground-based natural covers have the same form factor than road). Finally, the mean radiant temperature for the person in the sun is expressed according to the expression:

$$T_{mrt} = \sqrt[4]{\frac{a_b}{\sigma\epsilon_b}\left(\frac{f_p}{sin(\gamma)}S^\downarrow + \Psi_{bs}S^\Downarrow + \Psi_{bw}S_w^r + \delta_r\psi_{br}S_r^r + \delta_g\psi_{br}S_g^r + \Psi_{bs}L^\downarrow + \psi_{bw}L_w^\uparrow + \delta_r\psi_{br}L_r^\uparrow + \delta_g\psi_{br}L_g^\uparrow\right)} \tag{21}$$

In this expression, the human body is characterized by a prescribed solar absorption coefficient ($a_b = 0.70$) and a prescribed emissivity ($\epsilon_b = 0.97$). In case the person is in the shadow, the term relative to direct short-wave radiation contribution is not taken into account, so that the expression becomes:

$$T_{mrt} = \sqrt[4]{\frac{a_b}{\sigma\epsilon_b}\left(\Psi_{bs}S^\Downarrow + \Psi_{bw}S_w^r + \delta_r\psi_{br}S_r^r + \delta_g\psi_{br}S_g^r + \Psi_{bs}L^\downarrow + \psi_{bw}L_w^\uparrow + \delta_r\psi_{br}L_r^\uparrow + \delta_g\psi_{br}L_g^\uparrow\right)} \tag{22}$$



## 5.2 Inclusion of tree effects

The new parameterization for trees in the TEB model requires to adapt the radiative contributions in the UTCI calculations for outdoor conditions. First, the initial contributions in diffuse short-wave radiation and infrared radiation (detailed in Eq. 21) are corrected for the sky and the walls from the radiation attenuation effect through the tree foliage. These attenuation coefficients

are those already presented in Redon et al. (2017) that are dependent on the leaf area density profile; they are referred to as $\tau_{sr}$ for attenuation between sky and road and $\tau_{wr}$ for attenuation between wall and road (see Eq. B1-B4 in Redon et al., 2017).

In addition, the tree foliage contributes himself to the total infrared flux received by the person, due to is own infrared emission (function of its temperature and its emissivity) and to the reemission of infrared radiation that it receives from ground and walls. One can note that Redon et al. (2017) made the hypothesis that the short-wave radiation received by tree foliage is

only reemitted upward. As a result, no contribution in diffuse short-wave radiation from trees is involved in the mean radiant temperature calculation. The final expression for a person in the sun is as following:

$$T_{mrt} = \sqrt[4]{\frac{a_b}{\sigma\epsilon_b}\left(\frac{f_p}{sin(\gamma)}S^\downarrow + \tau_{sr}\Psi_{bs}S^\Downarrow + \tau_{wr}\Psi_{bw}S_w^r + \delta_r\psi_{br}S_r^r + \delta_g\psi_{br}S_g^r + \tau_{sr}\Psi_{bs}L^\downarrow + \tau_{wr}\psi_{bw}L_w^\uparrow + \delta_r\psi_{br}L_r^\uparrow + \delta_g\psi_{br}L_g^\uparrow + \psi_{bt}L_t^\uparrow\right)} \quad (23)$$

The infrared emission contribution due to the the tree foliage is expressed as:

$$L_t^\uparrow = (1-\tau_{sr})\left\{\sigma\epsilon_t T_t^4 + (1-\epsilon_t)\Psi_{tr}(\delta_r L_r^\uparrow + \delta_g L_g^\uparrow) + (1-\epsilon_t)(1-\Psi_{tr})L_w^\uparrow\right\} \quad (24)$$

with $\epsilon_t$ and $T_t$ the emissivity and temperature of the tree foliage, respectively, and $\Psi_{tr}$ the form factor for tree regarding emission coming from the road and ground-based natural covers.

## 6 Evaluation of vegetated-canyon microclimate modelling under semi-arid conditions

An evaluation exercise of the TEB performances in simulating microclimatic conditions of an urban canyon with trees is performed for a real

study case, i.e. the experimental campaign presented by Shashua-Bar et al. (2009). The TEB model here applied includes the ensemble of developments relative to urban trees: (1) the radiative processes detailed by Redon et al. (2017) and already evaluated by comparison with a reference model, (2) the energy exchanges between trees and air volume within the canyon, and (3) the aerodynamic effect of trees on local airflow. This version is referred to as TEB-Tree and is compared to the reference version of the model TEB-Ref (Lemonsu et al., 2012) which already integrated natural covers within the canyon but like a ground-based located compartment.

## 6.1 Study area and experimental data

The experimental data have been collected on campus of Sde-Boqer in the semi-desert region of Negev of southern Israel (30.85°N, 34.78°E, 475 m of altitude) during summer 2007 (Shashua-Bar et al., 2009). Two semi-adjacent courtyards with comparable characteristics in terms of geometry and materials have been set-up according to six landscape arrangments incorporating various combinations of bare soil, lawn, and trees. The present study case concentrates exclusively on the arrangement with bare soil and trees. In this case the trees are a *Prosopis*

*juliflora* and a *Tipuana tipu*, that are common species for the region and are known to be water-consumption saving (Kremmer et Galon, 1996). A drip irrigation was installed for each tree around the trunk.





The courtyard was equipped with sensors recording (1) air temperature, relative humidity, vapour pressure and wind speed at 1.5 m above the ground, (2) radiation fluxes (incoming and outgoing radiation, and net radiation) at the roof top, (3) surface temperatures of eastern, western, and southfacing walls, pavement, soil and tree foliage, and (4) transpiration from the trees (using the sap flow method). A meteorological station located 400 m northwest of the site in an open desert area recorded for the same period air temperature and humidity at 1.5 m above the ground, wind speed and direction at 10 m, as well as soil temperature.

## 6.2 Numerical configuration and experiments

The simulations are performed by running TEB on a single grid point to which are attributed the descriptive parameters of the experimented courtyard considering its configuration is close to the concept of urban canyon applied in TEB. The TEB model input parameters are prescribed according to the detailed description of the site proposed by Shashua-Bar et al. (2009). The semi-enclosed courtyard is oriented with an angle of 12 ° from the north (clockwise). Its width is 5.5 m and it is bordered by two rows of three-meters tall buildings with flat roofs. These buildings are made of light concrete, as well as the pavement made of thin layers of light concrete laying on the ground. All impervious covers are light color with high albedo of 0.35 for walls and 0.40 for roofs and pavement. All thermal and radiative properties are listed in Table 1 in Lemonsu et al. (2012). The ground inside of the courtyard consist in 70 % of pavement and 30 % of bare soil. The tree crowns present a overlapping rate of 70 % of ground-based surfaces.

The TEB input parameters derived from these real data differ depending on the version TEB-Ref or TEB-Tree that is used (see the description of configurations in Figure 2 and Table 1). The TEB-Ref version treats the ensemble of natural elements like a composite cover. If there is trees, the spatial coverage of crowns is considered as a ground-based surface which can consequently mask other elements. This is the case here since the tree foliage covers 70 % of the canyon. So that in this configuration, the canyon in the TEB approach is described as being composed of 30 % of pavement and 70 % of nature. This fraction of nature is exclusively high vegetation in this case, whereas the fractions of low vegetation and of bare soil are 0. The TEB-Tree version describes more realistically the arrangement of elements by dissociating the tree-foliage stratum from ground-based natural covers. As a consequence in this case, the ground-based surfaces are the same than the real ones i.e. 70 % of pavement and 30 % of nature, this latter is exclusivement bare soil. In addition, the tree-foliage overlapping fraction is 70 % of the canyon.

The meteorological forcing data that must be provided to TEB, i.e. air temperature, humidity, wind speed, incoming short- and long-wave radiation, and atmospheric pressure above the top of the canyon, are coming from in situ measurements collected above the roof and from data recorded by the reference meteorological station. The method is described in details in Lemonsu et al. (2012).

## 6.3 Results of evaluation

The comparison of microclimatic conditions and surface temperatures observed and simulated within the canyon indicates an improvement of simulation results with the new TEB-Tree version in comparison with TEB-Ref (Figure 3). For air temperature, the improvement brought by TEB-Tree is noted for daytime hours: the simulated diurnal cycle is in better agreement with the observed one. In comparison, the TEB-Ref case indicates a too early warming in the morning. The statistical scores for the whole diurnal cycle (compiled in Table 2) are slightly better with a mean absolute error (MAE) and a root-mean square error (RMSE) of 0.69 and 0.89 ° C, respectively, instead 0.73 and 0.93 ° C for TEB-Ref. Inversely, a more important underestimation of air humidity by TEB-Tree than TEB-Ref is noted during the day, whereas an





increase of humidity could reasonably be expected in the new version considering the latent heat flux from natural covers is now vertically distributed in TEB-Tree. The reason of this bias is discussed in the analysis of vertical profiles presented in Section 6.4. A clear impact of the new parameterization is observed on wind. By considering the drag effect of tree foliage stratum in equations of TEB-SBL, the air flow is significantly decelerated. The wind speed simulated in TEB-Tree is significantly lower than in TEB-REF, that is more conform to

measurements. MAE et RMSE are reduced from 0.97 and 1.10 m s$^{-1}$, respectively, to 0.47 and 0.53 m s$^{-1}$. Nonetheless, an overestimation of wind persists during the day.

The surface temperatures, that is for walls or pavement, are significantly improved by taking into account the radiative effects of tree foliage (Figure 3). The incoming radiation received by the canyon facets is reduced due to interception and attenuation by trees that are so

tall than buildings. As a result, the surface temperature maxima are lower in TEB-Tree than in TEB-Ref by almost 10°C for eastern wall and pavement, a little bit less for western wall. Inversely at night, the tree foliage limits the cooling by trapping of infrared emission so that the surface temperatures are higher in TEB-Tree than TEB-Ref. The comparison to measurements confirms that all these modifications lead to an improvement in simulation of surface temperatures. TEB-Tree still overestimates the surface temperatures at daytime, and underestimates them at nighttime, compared to observations, suggesting the attenuation effect by tree foliage remains underestimated by the model in this

case.

Finally, the tree foliage temperature and the evapotranspiration of trees are analyzed (Figure 4). For TEB-Tree, the hypothesis is done that the tree foliage temperature is in equilibrium with air temperature. The comparison to measurements shows that in the present case, the foliage is a bit warmer than ambiant air by around 2 ˚C during the day. The tree transpiration flux is correctly simulated by the model.

Especially, the daily maximum is more realistic in TEB-Tree than TEB-Ref. The transpiration is indeed overestimated by TEB-Ref during daytime hours probably due to the too strong wind. The new parameterization makes possible to significantly reduce this bias (Table 2). Nonetheless, a default persists in the morning. According to the measurements conducted with the sap-flow method, the transpiration process starts very early (about 5 am) whereas it is delayed in the simulations by almost two hours. The origin of this bais is not clearly identified: it could be explained by the parameterization of transpiration itself but the sensitivity tests done did not make possible to fix the problem.

The sap-flow measurements highlight that the transpiration flux starts before sunrise which the model is not able to simulate (as long as net radiation is zero).

## 6.4 Impact of mixing length on atmospheric vertical profiles

The mixing length ($L$) is parameterized in TEB on the basis of the work of Santiago and Martilli (2010) according to the height of buildings, the frontal area density, and the displacement height, i.e. parameters depending on the geometry of the canyon (see Eq. 10-12 in Lemonsu

et al., 2012). However, the formulation depends on the vertical level considered within or above the urban canopy layer. In the lower half of the canyon, the mixing length (which characterizes the size of turbulent eddies) is constrained by the distance to the ground.

Nevertheless, trees can be expected to influence the size of eddies in the canyon. Therefore, a formulation which takes into account the presence of trees in a simple way is tested here in order to assess the impact on microclimatic variables in the canyon. It is simply assumed

that (1) within the foliage layer, the mixing length is arbitrarily fixed to 10 cm; and (2) below the trees, the mixing length ($L_t$) is constrained by the minimum distance between the ground distance and the distance to the base of the tree-foliage layer. A mean mixing length ($L_{can}$) is then calculated as the average of the two mixing lengths for the treeless part of the canyon ($L$) and the part with trees ($L_t$) according to the



tree cover fraction:

$$L_{can}(z) = (1 - \delta_t)\, L(z) + \delta_t L_t(z) = (1 - \delta_t)\, L(z) + \delta_t\, min\{z, h_{tk} - z\} \tag{25}$$

Figure 5 compares the vertical profiles of air temperature, specific humidity, and wind speed simulated in the canyon at 3 am and 3 pm by TEB-Ref, TEB-Tree, and TEB-Tree($L_t$) which considers the new mixing length parameterization depending on trees. Day and night, TEB-Ref simulates a quite progressive decrease in wind speed when approaching the ground, due to the drag effect of walls over the entire height of canyon. For TEB-Tree, the additional drag effect of trees leads to a strong inflection on wind speed between the 3rd and 4th vertical levels of TEB-SBL i.e. where the foliage layer is located. This inflection is still reinforced with TEB-Tree($L_t$) so that the vertical profile is shifted towards slightly lower wind speeds within the canyon that result in better statistical scores compared to observations: MAE and RMSE of 0.28 and 0.32 m s$^{-1}$, respectively, instead 0.47 and 0.53 m s$^{-1}$ for TEB-Tree (Table 2).

During the day, TEB-Ref simulates a specific humidity profile that increases sharply as it approaches the ground. In this version, all vegetation is simulated on the ground and covers the bare soil and a part of pavement. Therefore, the evaporation term (through the latent heat flux) is strong and is injected into TEB-SBL at the first vertical level, which explains the shape of the simulated profile. The TEB-Tree humidity profile is more homogeneous because the moisture supply comes from ground for part, but also from the trees in the levels above. But the air specific humidity remains overall lower than in TEB-Ref because the real cover fraction of vegetation is lower. TEB-Tree($L_t$) reduces turbulent mixing below the trees which tends to trap moisture longer in the lower layers. This partially correct the biases of TEB-Tree by improving significantly the scores (see Table 2).

Finally for temperature, TEB-Ref simulates a progressive decrease in temperature in the canyon because the configuration with an aspect ratio of 0.55 does not result in much radiative trapping. On contrary, the foliage layer of TEB-Tree blocks infrared emissions from walls and floors within the canyon while being itself a significant source of infrared emissions at night. This explains why the temperature is significantly higher than in TEB-Ref. During the day, TEB-Ref simulates an homogeneous profile in the canyon because the heat contributions (in the form of sensible heat flux) come from both ground-based surfaces and walls. TEB-Tree shows a very substantial decrease in temperature because the trees strongly reduce the radiation received by the ground and the walls. As a result, surface temperatures are lower (as noted in Figure 3) and sensible heat fluxes are reduced. By reducing turbulent mixing, TEB-Tree($L_t$) does not evacuate heat outside the canyon and gives a very different profile from TEB-Tree with much warmer air temperatures in the canyon.

### 6.5 Modelling of thermal comfort

An interest of this new parameterization for trees in urban canyons is to better predict outdoor thermal comfort conditions. The air temperature is not so different between the two experiments TEB-Ref and TEB-Tree. The modifications in radiation exchanges, energy fluxes, and ventilation induced by the presence of trees may have nonetheless a significant impact on heat perception by people.

This is here quantified through the UTCI diagnosis whose formulation has been adapted in TEB-tree in order to include the radiation effects due to tree-foliage stratum as detailed in Section 5.2. Even if no observation of thermal comfort is available for an objective evaluation, it is however interesting to compared the UTCI simulated according to both versions TEB-Ref and TEB-Tree. These results are presented In Figure 6 for the situation for which people are in the sun (left) or in the shade (right). For indicative purpose, the air temperature simulated in the canyon is also presented. The UTCI for cases in the sun and in the shade only differ for daytime hours because at night radiation





conditions are the same. For TEB-Ref, the UTCI is greater than air temperature between 6 am and 3 pm for the case in the sun, and then becomes largely lower than air temperature by 6-7 ° C during the evening and night. This high daily amplitude is mainly driven by the radiative exchanges. The UTCI calculated in the shade is significantly lower than that in the sun during daytime: people are more preserved from heat due to shadow effects. It remains in this case always below air temperature. Using TEB-Tree, the UTCI is quite different. During the day,

it is slightly greater than for TEB-Ref because the wind speed is weaker due to the drag effect of trees. But the main difference is noticed during the night (and in a lesser extend, in the morning and in the evening): the infrared radiation downward emitted by the tree-foliage stratum - potentially received by a person in the street - significantly limits the decrease in nocturnal UTCI. It remains 5 ° C than in TEB-Ref simulations. This result is important to be emphasized which highlights that street trees may degrade thermal comfort conditions at night by trapping radiation and amplifying downward infrared emissions inside the canyon while reducing ventilation.

**7   Conclusions**

The TEB model has evolved considerably in recent years, particularly with the objective of improving the representation of vegetation in urban areas. After the implementation of ground-based low vegetation inside the canyon, and of green roofs on buildings, an explicit representation of the tree-foliage layer was implemented. A detailed parameterization of associated radiative processes was developed and tested by Redon et al. (2017) with very encouraging results. The whole issue in this study was to treat the energy exchanges of vegetation by taking

into account the dissociation of vegetation strata, and to include the drag effect of trees on wind speed in the canyon.

The strategy is rather simple by maintaining the bigleaf approach for using the ISBA SVAT model i.e. by calculating a single energy balance for natural covers, treated as a composite compartment. Nevertheless, the incident radiative fluxes provided to ISBA are calculated as a weighted average of those received by the natural ground surfaces and those received by the trees according to the coverage fractions. The

energy fluxes calculated by ISBA are then redistributed according to the same principle between ground surfaces and trees, then injected into the SBL parameterization of TEB for the calculation of the air temperature and humidity vertical profiles of the air in the canyon. Besides, a drag term of trees depending on the horizontal coverage fraction and the leaf density profile of trees is included in evolution equations of momentum and TKE. It is in addition to the drag term already configured for buildings.

An evaluation exercise conducted in comparison to field measurements shows that thanks to these new developments, the model better simulates surface temperatures and air temperature in the canyon. The main improvement concerns the wind, which is now slowed down by the presence of trees as observed. Some tests were also dedicated to the formulation of a new mixing length. They showed a significant sensitivity in temperature and humidity vertical profiles modelling to these parameterization choices. However, more complete evaluations are required to set the parameterization in a more robust and objective way. Also in the future, it could be necessary to move towards a

Multi-Energy Budget (MEB) approach (Boone et al., 2017) to solve the energy balance separately from ground surfaces and trees. The bigleaf approach reveals limitations, and raises conceptual questions about the choice of forcing level for energy fluxes calculations or the representativeness of a single temperature attributed to the composite layer of natural covers.

Finally, the UTCI diagnosis was reformulated by taking into account the presence of trees. This essentially consisted of adding infrared

emissions from trees to the mean radiant temperature calculation. The other modifications presented here and in Redon et al. (2017) have also an influence on the UTCI because they modify radiative exchanges, and micrometeorological variables in the canyon, particularly wind.





Although it was not possible here to evaluate the simulated UTCI due to lack of observation, it is very interesting to note that the presence of trees degrades the thermal comfort conditions during the night. Additional evaluations were conducted for another study site (de Munck et al., 2018) and confirmed this effect. This underlines the relevance of explicitly taking trees into account in urban climate models in order to more realistically model urban design strategies and impacts on comfort.

5 *Code availability.* The TEB code is available in open source via the surface modeling platform SURFEX, downloadable at http://www.umr-cnrm.fr/surfex/. This Open-SURFEX will be updated at relatively low frequency (every 3 to 6 months) and developments presented here are not yet included in the last version. If you need more frequent updates, or if you need what is not in Open-SURFEX (DrHOOK, FA/LFI formats, GAUSSIAN grid), we invite you to follow the procedure to open a GIT account and to access real-time modifications of the code (see instructions in the previous link).

10 *Data availability.* The model outputs are available upon request from the corresponding author. The experimental data that are used for the evaluation stage were provided by Prof. Evyatar Erell from Ben-Gurion University of the Negev. To access this data, it is necessary to contact directly Prof. E. Erell.

*Author contributions.* All three co-authors have contributed to the development and improvement of the TEB's code in SURFEX V8.0. Besides, E. Redon and A. Lemonsu performed the evaluation step by performing the simulation and comparing the model results with the 15 experimental data.

*Acknowledgements.* We acknowledge prof. Evyatar Erell from Ben-Gurion University of the Negev for giving us access to the experimental data.



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





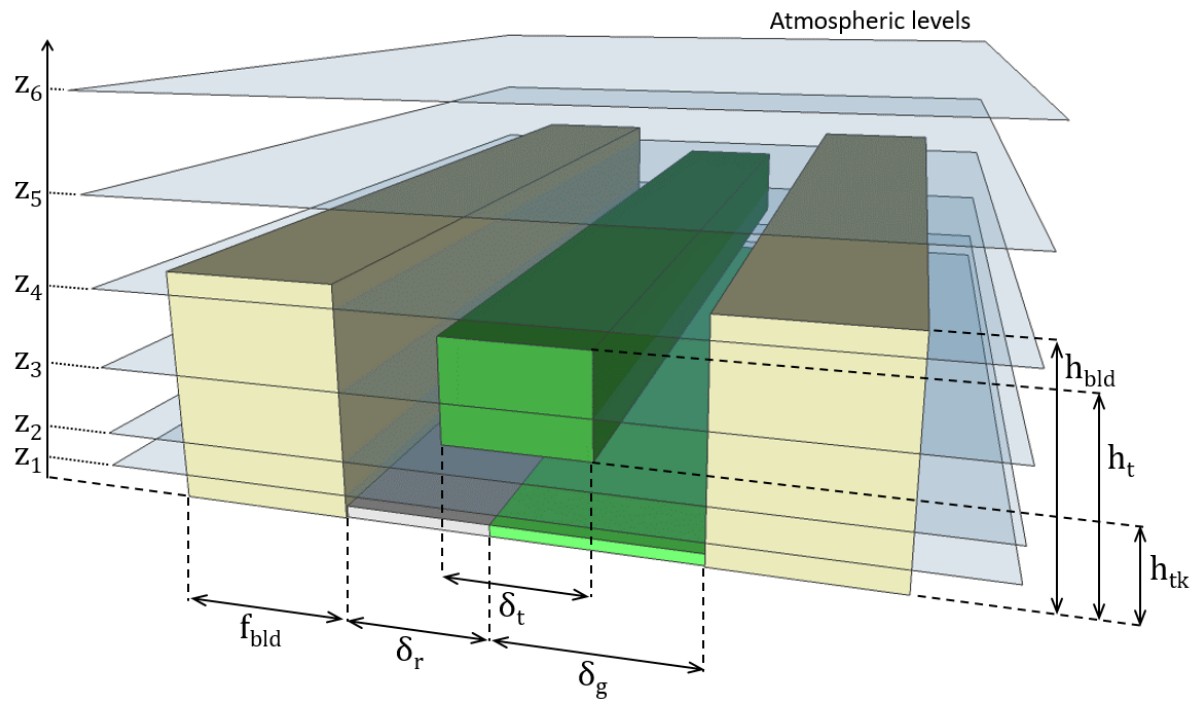

**Figure 1.** Schematic representation of the TEB's urban canyon integrating a part of ground-based natural covers and an explicit tree-foliage layer, and of the atmospheric vertical levels of the SBL scheme coupled to TEB to compute vertical profiles of micrometeorological variables inside and above the canyon.





(a) TEB-Ref configuration          (b) TEB-Tree configuration

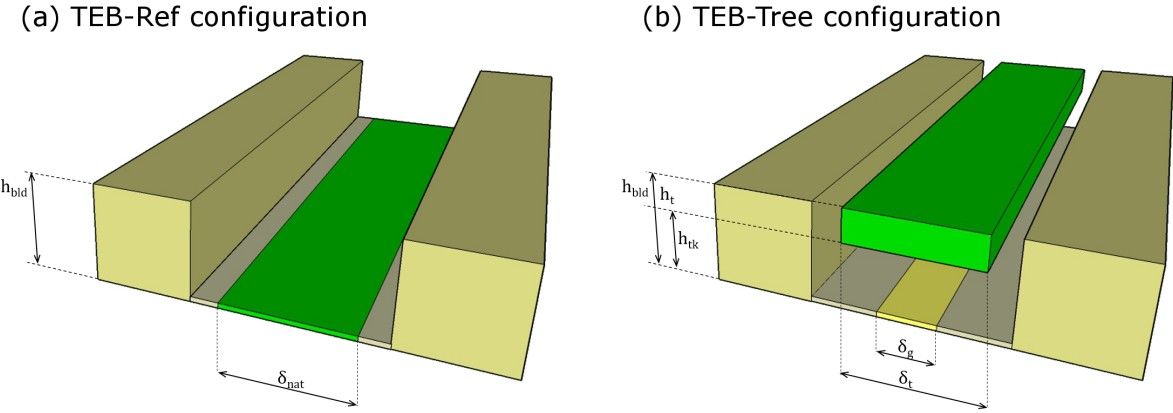

**Figure 2.** Comparison of the two urban canyon configurations prescribed to represent the experimental site of Sde-Boqer according to the TEB-Ref (a) and TEB-Tree (b) versions.

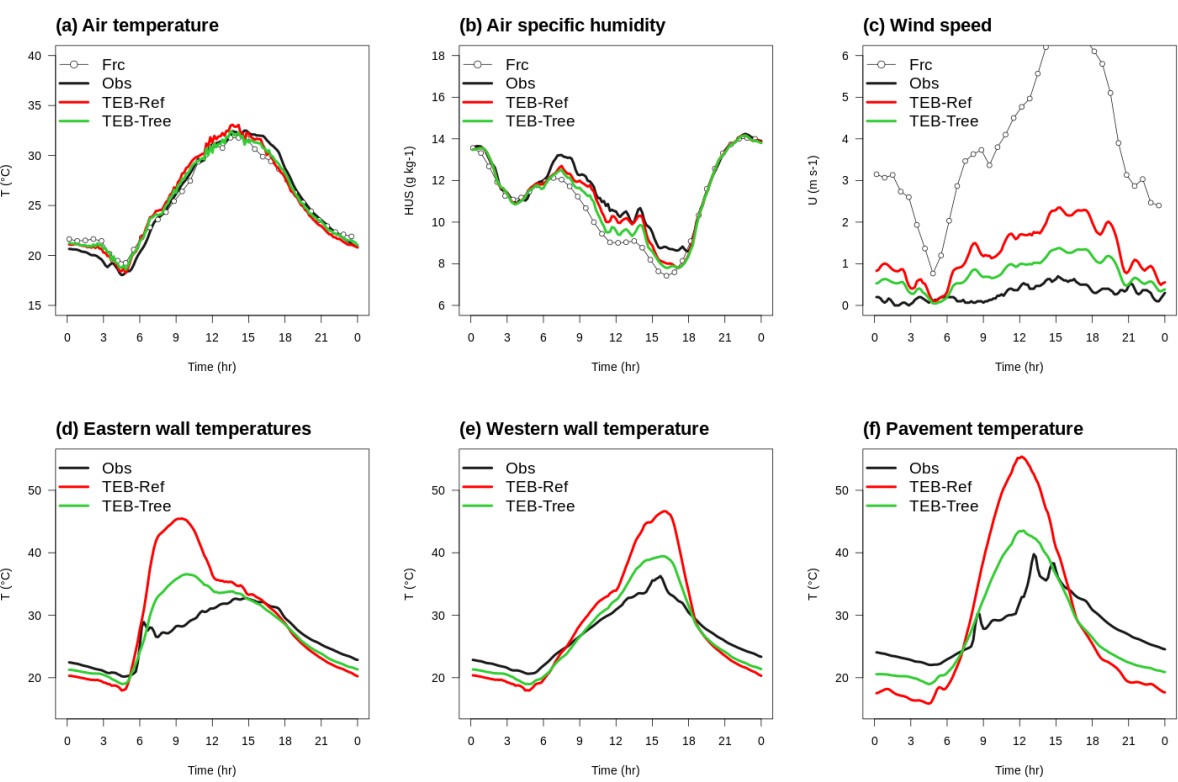

**Figure 3.** Comparison of the TEB-Ref and TEB-Tree results (red and green lines, respectively) with meteorological variables (top panel) and surface temperatures of urban facets (bottom panel) measured within the courtyard (black line). For air temperature, specific humidity and wind speed, the gray line with symbols indicates the forcing data above the buildings.





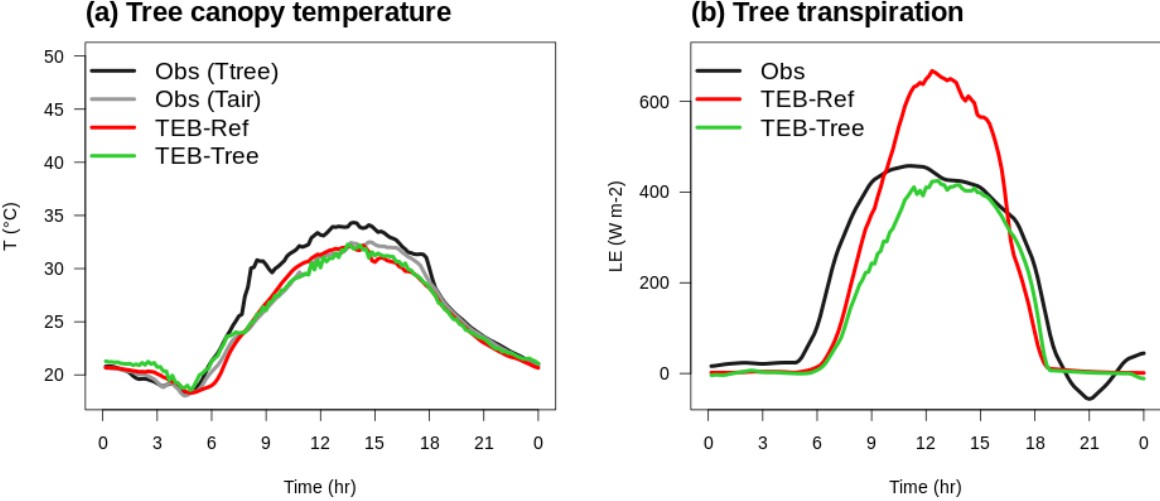

**Figure 4.** Comparison of the TEB-Ref and TEB-Tree results (red and green lines, respectively) with foliage temperature and tree transpiration flux measured for trees located within the courtyard (black line).

**Figure 5.** Comparison of vertical profiles in air temperature, specific humidity, and wind speed simulated within the courtyard according with TEB-Ref (red line), TEB-Tree (green line), and TEB-Tree including a new parameterization for mixing length (dashed green line). The location of tree-foliage layer is represented by the hatched strip, and the measurement level by the dashed black line.





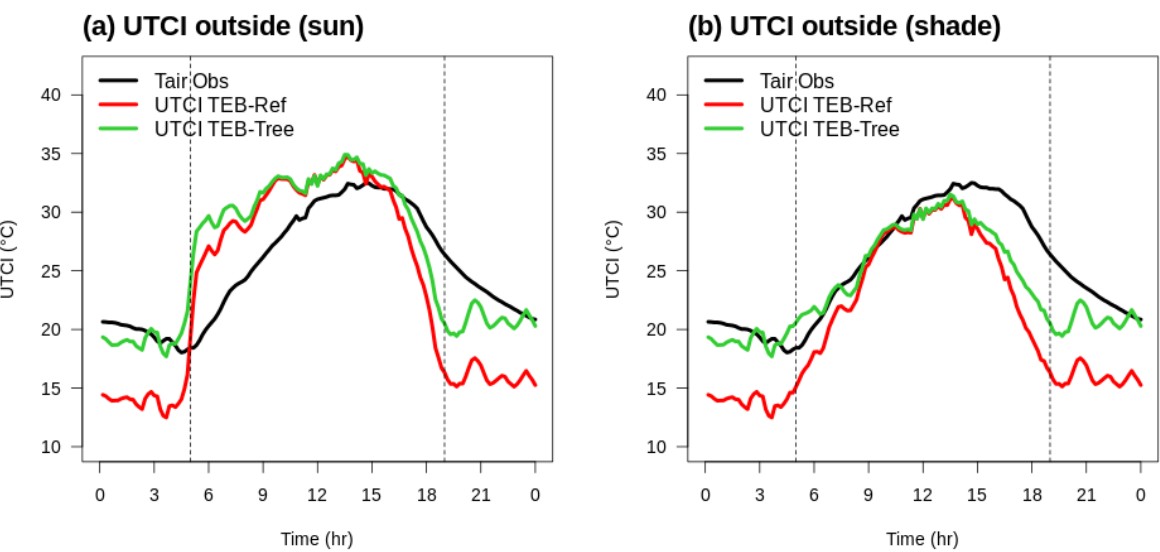

**Figure 6.** Comparison of UTCI in the sun (a) and in the shade (b) simulated by TEB-Ref (red line) and TEB-Tree (green line). The black line is the air temperature measured within the courtyard.





**Table 1.** TEB's input parameters according to the two configurations prescribed to represent the experimental site of Sde-Boqer according to the TEB-Ref and TEB-Tree versions (in accordance with Figure 2).

|  |  | *TEB-Ref* | *TEB-Tree* |
|---|---|---|---|
| Building fraction | (-) | 0.35 | 0.35 |
| Pavement fraction | (-) | 0.20 | 0.45 |
| Ground-based nature fraction | (-) | 0.45 | 0.20 |
| - High vegetation | (-) | 1.00 | 0.00 |
| - Low vegetation fraction | (-) | 0.00 | 0.00 |
| - Bare soil fraction | (-) | 0.00 | 1.00 |
| Tree overlapping fraction / canyon | (-) | 0.00 | 0.70 |
| Building height | (m) | 3.0 | 3.0 |
| Aerodynamic roughness length | (m) | 0.3 | 0.3 |
| Wall-plan area ratio | (-) | 0.71 | 0.71 |
| Canyon aspect ratio | (-) | 0.55 | 0.55 |





**Table 2.** Mean absolute error (Model-Obs) and root-mean square error in temperature, humidity, and wind speed at 1.5 m agl, in surface temperature of walls and tree foliage, and in evaporation. The scores are calculated for TEB-Ref, TEB-Tree, and TEB-Tree including a new parameterization for mixing length.

| | | *TEB-Ref* | | *TEB-Tree* | | *TEB-Tree ($L_t$)* | |
|---|---|---|---|---|---|---|---|
| | | MAE | RMSE | MAE | RMSE | MAE | RMSE |
| $T_{1.5m}$ | (°C) | 0.80 | 0.96 | 0.69 | 0.89 | 0.80 | 1.01 |
| $q_{1.5m}$ | (g kg$^{-1}$) | 0.29 | 0.41 | 0.45 | 0.60 | 0.39 | 0.51 |
| $U_{1.5m}$ | (m s$^{-1}$) | 0.98 | 1.11 | 0.47 | 0.53 | 0.28 | 0.32 |
| $T_{Swall_{(East)}}$ | (°C) | 4.65 | 7.08 | 2.32 | 3.26 | 2.62 | 3.68 |
| $T_{Swall_{(West)}}$ | (°C) | 3.78 | 4.96 | 1.91 | 2.27 | 2.23 | 2.68 |
| $T_{Spavement}$ | (°C) | 8.10 | 9.97 | 4.14 | 4.96 | 4.47 | 5.42 |
| $T_{Tree}$ | (°C) | 1.45 | 1.87 | 1.52 | 2.01 | 1.36 | 1.81 |
| $LE_{Tree}$ | (°C) | 86.47 | 110.89 | 59.37 | 83.89 | 77.74 | 99.79 |