# Peer review of "An urban trees parameterization for modelling microclimatic variables and thermal comfort conditions at street level with the Town Energy Balance model (TEB-SURFEX v8.0)"

_Geoscientific Model Development, 2019_

## Referee Comment (RC1) · Anonymous Referee #1 · 28 Jun 2019

This paper presents a new urban tree parameterization for a urban canopy model (TEB). While there have been several tree parameterization schemes in the field, this study clearly pushes the modeling forward by incorporating ground vegetation-tree interactions. Overall the paper is well organized and written, but some modeling details are not clear enough.

Specific comments: 1) Section 2.2: the authors refer to Redon et al. (2017) for the radiative effects of urban trees. However, no ground vegetation is considered in Redon et al. (2017), so how do the trees modify net radiation for underlying ground vegetation

(e.g., Fig. 1). The authors should present the surface temperature of ground vegetation at some time to illustrate the impact of trees. Please provide more information.

2) Section 4.1: Equation (10) accounts for the drag force of urban canyon in both horizontal and vertical directions. If I understand correctly, as shown in Fig.1, TEB model assumes a very long canyon as compared to the building width, which leads to large variations of vertical drags when wind directions change. Is the frontal index of buildings/trees considered in the model? Please elaborate.

3) Section 4.2: How is the transpiration rate of trees calculated? It seems that Equation 8 is missing in the paper. I love the consideration of leaf area density as a vertical profile to advance the ET modeling of trees. But why would the authors use a constant LAD then? At least some results with vertically varied LAD should be presented, especially in terms of the transpiration rate.

4) Section 5.2: The attenuation coefficients are sensitive to the locations of the trees and pedestrian. What are the locations of trees and pedestrian in this case? Also the authors separate trunks from the crown. Should the radiation from trunks be considered in this case?

I would like to comment more, especially when you have non-cylinder shape of crowns the mixing length below the trees may also be modified. But at this moment without sufficient details of the parameterization it is hard to assess the results.

Editorial comments: Abstract: ISBA model should be spell out at its first mention

---

## Referee Comment (RC2) · Anonymous Referee #2 · 16 Jul 2019

Overall:

The present paper details an update to the TEB urban canopy model. It demonstrates that the updates made improve comparisons against observations made in an arid climate. It further demonstrates that in warm climates, trees may degrade thermal comfort at night.

The authors are encouraged to compare their updated model not only to their original model, but also contextualize their modelling approach relative to other urban canopy

tree models, e.g. in a Discussion section. How does the current work differ, and what are its advantages and disadvantages? In particular, it is not clear how the current approach offers conceptual and/or operational advantages over other approaches cited (lines 2.28-29), even though it is shown that the current approach is better than the previous one with ground vegetation only. Potential error introduced by combining tree and ground vegetation energy balances is not assessed.

In general, some more discussion of the results presented in the figures, especially Fig. 3, would be helpful in illustrating TEB-tree's usefulness and novelty. For example, there are some variables for which the difference between observations and model output is still appreciable and may require elucidation. Other model outputs do not appear to be realistic (e.g., daytime UTCI in the sun is not reduced with TEB-Tree, which should include impacts of tree shade).

Finally, this paper would benefit from editing for grammar and language.

Specific comments:

2.8: Is scattering accounted for?

The introduction provides an acceptable overview and motivation. The distinction between models that resolve vegetation and "urban canopy models" is not made fully clear – in both cases the scale of vegetation elements is smaller than the model grid.

4.1: How would representing the sides of the crown help?

4.28: How are these parameters defined?

Eq. 4 & 5: These fluxes appear to be defined per plan area fraction. How is leaf area taken into consideration? A particular coverage of trees can have higher or lower leaf area.

Eqs. 6 & 7: It is not clear that this approach can be expected to give accurate results, as we have no reason to expect that QH and QE scale linearly with vegetation fraction at different levels. Stomatal resistances are likely to differ strongly between trees and ground-based vegetation, as are aerodynamic resistances (trees are likely to be exposed to higher winds). As well, how is storage heat flux treated here, given that soil will store a lot more heat than trees?

5.26: Where is equation 8?

5.27: Probably QEg and QEt is meant.

6.10: Which pressure gradient?

6.20-23: Eqs. 11-14 do not allow for the vertical transport of any of the quantities therein. Discussion of how vertical transport is treated should be included in these equations or better, in Eq. 9.

Eq. 11: What is the last term? The square of the friction velocity. How is the friction velocity defined in the urban canopy?

Eq. 15: It appears that LAD is assumed vertically uniform, and therefore turbulent fluxes from trees are assumed to be uniform with height in the tree canopy?

7.15: In Fig. 1 tree crowns are illustrated as uniform layers, not cylinders.

9.11: "only reflected upward" – shortwave is not emitted by trees.

9.12: Why is this re-emitted radiation only upward?

9.10-12: Is it possible to estimate the potential error from this assumption? How much could it affect the Tmrt?

10.2: Citation not in reference list.

Section 6.2: For air temperature and humidity surely this canyon is the wrong scale (too small) to evaluate TEB-Tree? Advective effects are likely to be large?

10.22-30: This paragraph is difficult to understand.

Fig. 1 & Fig. 2: The text appears to indicate that trees do not have sides in this model, but this diagram suggests the opposite.

Fig. 3 d & f: Some of the model-measurement surface temperature differences are very large (>10 K). What is the model missing in these cases? The modelled increase in east wall temperatures in the morning is not reproduced in observations.

11.20: "... by trees that are so tall than buildings". Do you mean trees that are as tall as buildings? Or that are taller than buildings? Can trees that are taller than buildings be implemented in this model?

11.28: Tree temperature = air temperature ?? It seemed earlier that an energy balance is solved for leaves/trees, which means a surface temperature is calculated?

12.13: Some justification into the selection of the mixing length would be useful. 10 cm as a mixing length seems somewhat low. Maybe scale the mixing length as LAD?

Eq. 25: What about the presence of buildings and their effects on mixing length? Is this already included here?

Fig.5: Would perhaps benefit from increased vertical resolution.

Section 6.4: Which version, with or without the length scale modification, do the authors think is best and consider to be "TEB-Tree"?

Fig. 6: Surely TEB-Tree, by virtue of accounting for shade from trees, should strongly reduce UTCI during daytime in the sun relative to TEB-Ref, which does not account for tree shade?

13.22: "at night radiation conditions are the same." What about longwave radiation? Surely that differs when near a wall at night? Do you mean to say that at night there is no shortwave radiation?

13.30: "It remains 5 C (warmer?) than in TEB-Ref simulations"

Section 6.5: Regarding the degradation of UTCI at night due to trees – are there measurements of this effect (or of increased longwave from urban tree cover) that you can reference, for example, to indicate that your results are in line with observations?

---

## Author Comment (AC2) · 1 Oct 2019

**An urban trees parameterization for modelling microclimatic variables and thermal comfort conditions at street level with the Town Energy Balance model (TEB-SURFEX v8.0)**
by Emilie Redon et al.

**Response to reviewers' comments**

**General response to all**

In response to the reviewers' comments, some important changes have been made to the new version of the manuscript:

1. Only the TREE-BARE configuration (courtyard with trees and bare soil) was presented in the previous version. To address the issue of taking into account both tree and herbaceous vegetation in TEB-Tree, we also simulated the TREE-GRASS configuration (courtyard with trees and lawn) for which we also had data. The manuscript now presents the evaluations (figs) for the TREE-GRASS case + the scores for both TREE-GRASS and TREE-BARE cases.

2. The section 6.4 on the new formulation of the mixing length raised many questions. It is true that we do not currently have adequate data to precisely evaluate this formulation and the potential benefit of the modification. Therefore, we preferred to delete this section since the formulation of mixing length remains unchanged in TEB-Tree compared to TEB-Ref.

3. The Section about the UTCI simulation has been modified to improve its clarity and relevance. The first part compares the UTCIs simulated by TEB-Ref and TEB-Tree to discuss the role of parameterization. A second part compares the different courtyard layouts and the impact on comfort conditions by putting these results in perspective with the works of Shashua-Bar et al (2011, Fig.2).

4. Finally, an error was found in the initialization of the material properties (wall albedo). Simulations were redone and figs and scores updated.

**Response to Referee #1**

**Specific comments:**

1) Section 2.2: the authors refer to Redon et al. (2017) for the radiative effects of urban trees. However, no ground vegetation is considered in Redon et al. (2017), so how do the trees modify net radiation for underlying ground vegetation (e.g., Fig. 1).

>> It is true that the comparisons TEB vs SOLENE carried out by Redon et al (2017) were based on simulations of vegetated canyons incorporating only tree vegetation. In this case, the ground surfaces were road. Nonetheless, the code has been developed to be able to include within the canyon a fraction of ground pervious covers that is defined as a combination of bare soil and herbaceous vegetation. This is explained is Redon et al. (2017), for instance, in Fig.1, in the descriptive parameters presented in Table 1, or the text in Section 3.3. In this paper, Eq. 14 expresses the solar radiation absorbed by the ground surface (same expression for road or natural covers) including the effect of trees.

The authors should present the surface temperature of ground vegetation at some time to illustrate the impact of trees. Please provide more information.

>> As explained in the article (Section 3.1), the model is based on the "bigleaf" approach for the SEB calculation. This implies that all natural covers (tree vegetation, herbaceous vegetation, and natural soils) are combined in a composite natural compartment. Then, the model resolves for this compartment a single SEB, as well as a single temperature evolution equation. This temperature is an average or composite temperature of the natural covers. Therefore, the temperature of the vegetation located on the ground under the trees cannot be analyzed separately.
As a complement, we present in Fig 3f both Tnat (composite temperature of natural covers) and Tt (tree temperature) and discuss the differences in Section 6.3.3.

2) Section 4.1: Equation (10) accounts for the drag force of urban canyon in both horizontal and vertical directions. If I understand correctly, as shown in Fig.1, TEB model assumes a very long canyon as compared to the building width, which leads to large variations of vertical drags when wind directions change. Is the frontal index of buildings/trees considered in the model? Please elaborate.

>> The wind profile is set in TEB as a mean horizontal wind profile that does not take into account the specific orientation of the streets. The drag coefficient associated with buildings is constant (and applied consistently to the vertical for wind speed calculation). The same is true for vegetation, except that this coefficient is weighted vertically, depending on the leaf density profile.
The term of frontal area density is used in TEB to calculate the mixing length, according to the formulations proposed by Santiago and Martilli (2010). This term is calculated as a mean frontal area density simply derived from the geometric properties defined in TEB (wall surface density) but without any link to wind direction. All this is described by Lemonsu et al (2012).
We clarified in the text:
*"The vertical transport of heat, humidity and momentum within and above the canyon is calculated by applying the turbulence scheme of Cuxart et al. (2000). This scheme is based on an equation for the turbulent kinetic energy, and is closed with a mixing length. This is parameterized in TEB on the basis of the work of (Santiago et al. 2010) according to the height of buildings, the mean frontal area density, and the displacement height, i.e. parameters depending on the geometry of the canyon (see Eq. 10-12 in Lemonsu et al. 2012)."*

3) Section 4.2: How is the transpiration rate of trees calculated?

>> As explained in Lemonsu et al. (2012) in Appendix B, ISBA calculates the latent heat flux as a combination of evaporation/transpiration terms produced by the natural covers: evapotranspiration from vegetation (**QEnat$_v$**), evaporation from the ground (**QEnat$_g$**) and from the ground with freezing (**QEnat$_{gi}$**), and vaporisation from snow (**QEnat$_s$**): **QEnat = QEnat$_v$ + QEnat$_g$ + QEnat$_{gi}$ + QEnat$_s$**

More specifically, the transpiration is calculated as: **QEnat$_v$ = ACnat f$_v$ L$_v$ (1− v$_{sn}$ ) Hv (qsat − qa )**

with **ACnat** aerodynamic conductance for heat transfer
$\quad$ **$f_v$** $\quad$ fraction of vegetation
$\quad$ **$v_{sn}$** $\quad$ fraction of vegetation covered by snow
$\quad$ **qsat** $\quad$ humidity at saturation
$\quad$ **qa** $\quad$ air humidity above the top of the canopy
$\quad$ **Hv** $\quad$ Halstead coefficient (i.e., the relative humidity of vegetation canopy as defined by Noilhan and Planton, 1989)
$\quad$ **$L_v$** $\quad$ vaporisation heat coefficient

>> Conforming to the "bigleaf" approach, the transpiration is calculated for the vegetation of the complete natural cover compartment. The vegetation can include only trees (as for experiment TREE-BARE), or combine trees and grass (as for experiment TREE-GRASS). In this last case, the transpiration is not calculated separately for trees and grass but for a composite vegetation.

It seems that Equation 8 is missing in the paper.

>> The label (8) that appears on page 5 is due to a layout problem. There is no missing equation here. The equations of turbulent latent heat fluxes are not presented because they are identical to those of sensitive heat fluxes (as inidcated in the text).
However, there was a typo: *"$Q_{Et}$ and $Q_{Et}$"* instead of *"$Q_{Eg}$ and $Q_{Et}$"* which we have corrected.

I love the consideration of leaf area density as a vertical profile to advance the ET modeling of trees. But why would the authors use a constant LAD then? At least some results with vertically varied LAD should be presented, especially in terms of the transpiration rate.

>> We use a constant LAD profile because we assume that trees are rectangular or cylindral in shape, which is quite realistic for pruned street trees. The model has the possibility to specify other crown shapes (conical or round shapes) but we do not have suitable observations to really evaluate the consideration of these different shapes. Considering the simplifying assumptions applied in the parameterizations for radiative, energetic, or aeraulic calculations, it seemed to us superfluous to go in this level of detail. We consider that remaining a homogeneous shape in the vertical is sufficient for now.

4) Section 5.2: The attenuation coefficients are sensitive to the locations of the trees and pedestrian. What are the locations of trees and pedestrian in this case? Also the authors separate trunks from the crown. Should the radiation from trunks be considered in this case?

>> This point refers to the attenuation coefficients of radiation by the trees foliage. This is an aspect that is discussed in more detail in Redon et al (2017), which focuses on radiative calculations, but little in this paper, which does not have this as its primary objective.
Again, it should be kept in mind that the model represents in a simplified and parameterized way the tree vegetation and its effects on radiative exchanges within the canyon:
1. Trunks are not taken into account as for of this kind of models. Only foliage layer is considered.
2. Trees are defined as a coverage fraction in the canyon, with an associated height and thickness for the foliage layer. But the actual positioning of the trees is not considered. For example, the model is not able to explicitly represent two separate rows of trees. This is well described in the article by Redon et al (2017).
3. For UTCI calculation, the model does not simulate the spatial variability of comfort conditions in the street. It simply calculates an "average" UTCI for an individual in the canyon by taking into account the radiative exchanges calculated on the basis of mean form factors.

I would like to comment more, especially when you have non-cylinder shape of crowns the mixing length below the trees may also be modified. But at this moment without sufficient details of the parameterization it is hard to assess the results.

**Editorial comments:** Abstract: ISBA model should be spell out at its first mention

>> This was corrected in the abstract for ISBA and TEB acronyms.

**Response to Referee #2**

**Overall:**

The present paper details an update to the TEB urban canopy model. It demonstrates that the updates made improve comparisons against observations made in an arid climate. It further demonstrates that in warm climates, trees may degrade thermal comfort at night.

The authors are encouraged to compare their updated model not only to their original model, but also contextualize their modelling approach relative to other urban canopy tree models, e.g. in a Discussion section. How does the current work differ, and what are its advantages and disadvantages? In particular, it is not clear how the current approach offers conceptual and/or operational advantages over other approaches cited (lines 2.28-29), even though it is shown that the current approach is better than the previous one with ground vegetation only. Potential error introduced by combining tree and ground vegetation energy balances is not assessed.

>> It seems to us very difficult to know how to compare objectively our results and approach to other models. Within the framework of radiative exchanges, we were able to compare TEB to a high-resolution architectural model. We constrained the modelling framework and certain parameters to focus on comparing the fluxes received, reflected and absorbed. For turbulent exchanges, thsi exercise is even more complex. In the conclusion, we highlight the limitations and issues related to the use of the bigleaf approach. We also mention the eventual need to move towards a multi-budget modelling for energy exchanges.

In general, some more discussion of the results presented in the figures, especially Fig.3, would be helpful in illustrating TEB-tree's usefulness and novelty. For example, there are some variables for which the difference between observations and model output is still appreciable and may require elucidation. Other model outputs do not appear to be realistic (e.g., daytime UTCI in the sun is not reduced with TEB-Tree, which should include impacts of tree shade).

>> We realize that the interpretation of the UTCIs presented in Fig. 6 is probably difficult for the reader to understand and that I need to clarify some points:
- The model independently calculates two UTCIs in the outdoor space i.e. the UTCI in the shade and the UTCI in the sun (but without taking into account the shading fractions in the street which can be more or less important depending on the configuration and layout of the canyon). This is why in the comparisons presented, there are relatively few differences on daytime UTCIs between the two model versions. The UTCI calculated in the sun does not take into account the shading effect of trees.
- To present a more understandable and realistic UTCI, I propose to calculate an "average" UTCI based on the UTCI in the sun and the UTCI in the shade, and weighted according to the canyon fractions in the sun and in the shade. Since the model does not explicitly represent the spatial location of the elements in the canyon (but simply coverage fractions), these shade/sun fractions are calculated simply as the ratio between the direct solar radiation received by the ground surface and the direct incident solar radiation at the top of the canyon.
- Based on this average UTCI, we can see more clearly the differences between the two versions of the model and the effect of tree shading on the UTCI during the day simulated by TEB-Tree. Figure 6 has therefore been updated and the discussion part of these results has been modified accordingly.

Finally, this paper would benefit from editing for grammar and language.

**Specific comments:**

- 2.8: Is scattering accounted for?

  >> This paragraph aims to describe how tree vegetation alters radiative exchanges in the canyon.
  We added a sentence: *"The scattering modulates the properties of reflected and transmitted radiation."*

- The introduction provides an acceptable overview and motivation. The distinction between models that resolve vegetation and "urban canopy models" is not made fully clear – in both cases the scale of vegetation elements is smaller than the model grid.

>> In the introduction, we have separated the presentation of CFD or radiative transfer models for which the shafts can be described as an assembly of voxels. Then, in a second step, we presented urban canopy models that rather adopt parametric approaches. This seemed to us enough detailed to introduce the scientific and objective questions of the present study.

- 4.1: How would representing the sides of the crown help?

Trees are seen as a flat horizontal surface (with an associated coverage fraction within the canyon) for calculating the interception of incoming solar radiation. But depending on position and thickness of the tree crowns, as well as inclination of the sun's rays, the edges of the crowns can intercept a part of this radiation. This effect is not taken into account in TEB, which leads to an underestimation of the incoming solar radiation interception (more or less important depending of canyon configuration and vegetation layouts). This limitation was underlined by Redon et al. 2017: *"... due to the expression of direct solar radiation intercepted by high vegetation at the top of the crown, which is treated as a horizontal surface in TEB (Eq. 1), the fluxes reaching the trees in TEB are globally underestimated compared to the SOLENE fluxes that include contributions on the vertical faces of the crown envelope "*

We modified and completed the text to clarified this point (Section 2.2):
*"Due to the simplified representation of trees geometry, a general defect is the underestimatation of the incoming solar radiation interception by the tree-foliage stratum. That is explained by the fact that TEB does not represent the sides of tree crowns, that can receive in reality a part of incoming radiation according to their position and inclination of the sun's rays."*

- 4.28: How are these parameters defined?

>> ISBA requires some parameters related to soil and vegetation i.e. Soil albedo, Vegetation and soil albedo, Vegetation and soil emissivity, Vegetation fraction, Stomatal resistance, Dynamical roughness length, Heat capacity of vegetation and of soil. In the standard approach where specific in situ data are not available and land covers are defined from the ECOCLIMAP database, default parameters are assigned according to vegetation types (ex. for trees, distinction is done between deciduous, evergreen trees, needleleaf evergreen trees, and for different boreal, temperate, tropical climate conditions).
For the evaluation exercise, these parameters were defined according to the local information available for the experimental site. We completed Section 3.1:
*"This compartment consists of fractions of bare soil, low vegetation, and high vegetation (and possibly snow). It is characterized by mean properties calculated from thermo-radiative (albedo, emissivity, heat capacity), aerodynamic (roughness length), and physiological (stomatal resistance for plants) parameters that are prescribed independently for the different types that compose it."*

- Eq. 4 & 5: These fluxes appear to be defined per plan area fraction. How is leaf area taken into consideration? A particular coverage of trees can have higher or lower leaf area.

>> The leaf area index is used to calculate the attenuation coefficient of radiation through the foliage layer (see Eq. 3, Redon et al. 2017). This has consequently an impact on the amount of radiation that passes through the tree canopy and reaches the natural ground-based surfaces (the term $S_g^\downarrow$). But in the expression of the fluxes weighting, we rely only on the horizontal coverage fractions.

- Eqs. 6 & 7: It is not clear that this approach can be expected to give accurate results, as we have no reason to expect that QH and QE scale linearly with vegetation fraction at different levels. Stomatal resistances are likely to differ strongly between trees and ground-based vegetation, as are aerodynamic resistances (trees are likely to be exposed to higher winds). As well, how is storage heat flux treated here, given that soil will store a lot more heat than trees?

>> You are absolutely right, it's a very strong assumption of parameterization. For now, we have preferred to keep the approach very simple rather than disaggregate the fluxes according to assumptions that are difficult to make. To go further, it would be necessary to drop the "bigleaf" approach and move towards a multi-energy budget calculation. Appropriate experimental data should also be available to assess the contribution of such an approach.
Concerning heat storage, the disaggregated fluxes between ground-based surfaces and trees are not diagnosed, simply because this is not necessary: this term does not come into play in the TEB-SBL parameterization.

- 5.26: Where is equation 8?

>> The label (8) that appears on page 5 is due to a layout problem. There is no missing equation here. The equations of turbulent latent heat fluxes are not presented because they are identical to those of sensitive heat fluxes (as inidcated in the text).
However, there was a typo: *"$Q_{Et}$ and $Q_{Et}$"* instead of *"$Q_{Eg}$ and $Q_{Et}$"* which we have corrected.

- 5.27: Probably QEg and QEt is meant.

>> Exactly, there was a typo: *"$Q_{Et}$ and $Q_{Et}$"* instead of *"$Q_{Eg}$ and $Q_{Et}$"* which we have corrected.

- 6.10: Which pressure gradient?

>> The pressure-gradient force is the force that results from a difference in atmospheric pressure and that comes into play in the geostrophic wind equation. This term is provided at the highest level of the TEB-SBL vertical grid by the atmospheric model when SURFEX is run in coupled mode, or assumed to be null in stand-alone configuration. This is already described and explained in the reference paper presenting the SBL parameterization (Masson and Seity 2007).

- 6.20-23: Eqs. 11-14 do not allow for the vertical transport of any of the quantities therein. Discussion of how vertical transport is treated should be included in these equations or better, in Eq. 9.

>> The vertical transport within and above the canyon is parameterised using the turbulent scheme of Cuxart et al. (2000) based on an equation for the turbulent kinetic energy, and it is closed with a mixing length. We clarified this pont in Section 4.1
*"The vertical transport of heat, humidity and momentum within and above the canyon is then calculated by applying the turbulence scheme of Cuxart et al. (2000). This scheme is based on an equation for the turbulent kinetic energy, and is closed with a mixing length. This is parameterized in TEB on the basis of the work of Santiago and Martilli (2010) according to the height of buildings, the mean frontal area density, and the displacement height, i.e. parameters depending on the geometry of the canyon (see Eq. 10-12 in Lemonsu et al., 2012)."*

- Eq. 11: What is the last term? The square of the friction velocity. How is the friction velocity defined in the urban canopy?

>> There was an error in the formulation. The friction velocity is decomposed according to the vertical levels of the SBL parameterization. So we replace u* by u*(k). We use u* at the ground for contribution of roads and u* at the top of buildings for contribution of roofs.

- Eq. 15: It appears that LAD is assumed vertically uniform, and therefore turbulent fluxes from trees are assumed to be uniform with height in the tree canopy?

>> As explained in Section 4.2, the turbulent fluxes of trees are distributed vertically according to LAD profile, i.e. homogeneously from the bottom to the top of trees, since LAD profile is assumed to be homogeneous for rectangular crowns.

- 7.15: In Fig. 1 tree crowns are illustrated as uniform layers, not cylinders.

>> The TEB model represents the canyon in a 2D plane with the assumption that this canyon has an infinite length (with the same shape and configuration). For canyons with trees, this implies that the crowns of the trees touch each other (without spacing between trees in the longitudinal direction of the canyon), creating a continuous layer of foliage.

We used the term of "cylinder" to characterize the tree crown shape. That was a way to refer to a standard type of pruned trees and to distinguish it from other shapes that may exist (ovoid or triangular), for which LAD profiles are different. But the term is indeed not well suited to the model.

For clarity, we changed cylindric by rectangular: *"For now, tree crowns are described rectangular in shape so that ..."*

- 9.11: "only reflected upward" – shortwave is not emitted by trees.

>> Yes you are right, it is corrected.

- 9.12: Why is this re-emitted radiation only upward?

>> It is assumed that it is mainly the top of the tree crowns that intercepts the incident solar radiation, so that this radiation is re-emitted upward. This formulation is the result of the work of Redon et al (2017) for the parameterization of radiative exchanges.

- 9.10-12: Is it possible to estimate the potential error from this assumption? How much could it affect the Tmrt?

>> This formulation proposed by Redon et al (2017) was applied in the comparison exercise to the high-resolution SOLENE model. It has been evaluated indirectly by evaluation multi-reflections within the canyon and solar radiation absorbed by the different elements of the canyon.

- 10.2: Citation not in reference list.

>> We now refer to Shashuabar et al. 2009.

- Section 6.2: For air temperature and humidity surely this canyon is the wrong scale (too small) to evaluate TEB-Tree? Advective effects are likely to be large?

>> It is indeed possible that there may be advection effects on the experimental site. This seems to be the case based on humidity measurements. This does represent a certain limitation for comparison to the model in forced mode on a grid point. Additional work is currently being carried out with a 3D numerical modelling configuration by coupling the surface scheme to a complete atmospheric model.

- 10.22-30: This paragraph is difficult to understand.

>> We modified the text to clarify: *"In the TEB-Ref standard approach, land cover fractions are calculated according to a single surface area without overlapping (as seen from the sky or by satellite). The sum of cover fractions is equal to 1. The tree cover has priority over what is below and is hidden, that is the case here since the tree-foliage layer covers 70 % of the canyon. In the TREE-BARE experiment, all bare ground and part of pavement are therefore hidden by trees, which modifies the real fractions (Table 1). In the TTREE-GRASS experiment, trees largely but not totally mask grass since the grass cover fraction is slightly greater than that of trees (Table 1). The TEB-Tree version describes more realistically the arrangement of elements by dissociating the tree-foliage stratum from ground-based natural covers. As a consequence, for both experiments, the cover fractions prescribed for TEB are the real ones (Table 1) and their sum is greater than 1."*

- Fig. 1 & Fig. 2: The text appears to indicate that trees do not have sides in this model, but this diagram suggests the opposite.

>> Indeed, the model does not treat the sides of the foliage layer. Nevertheless, it considers a base and a top for the foliage layer, as well as a coverage fraction in the canyon. The graphical representation

proposed in Figs. 1 and 2 therefore seemed to us to be the most explicit way of representing the model concept.

- Fig. 3 d & f: Some of the model-measurement surface temperature differences are very large (>10 K). What is the model missing in these cases? The modelled increase in east wall temperatures in the morning is not reproduced in observations.

>> Surface temperature simulations have been significantly improved by modifying the wall albedo whose prescribed value had been underestimated compared to the information from the experimental site. The albedo is now set at 0.6, which is high but corresponds to actual data provided by the team that made the measurements.
However, there is still a bias on the simulation of eastern wall temperature in the morning. I would say that this difference may be related to the specific arrangement of the trees in the canyon and the positioning of the sensor (which seems to be in the shade for a few hours). This effect is not found for the temperature of the west wall at the end of the day, which is correctly simulated by the model.
Figures and scores have be updated according to the new simulations. Additional comments have been included with regard to overestimation:
*"Nonethless despite the improvement, TEB-Tree still overestimates the surface temperature especially for eastern wall in the morning. This suggests the attenuation effect by tree foliage remains underestimated by the model in this case. The deviation from the measurement can also be explained by a difference between the compared quantities. The thermometer samples a specific area of the wall, while the model calculates an average temperature for the entire wall."*

- 11.20: ". . . by trees that are so tall than buildings". Do you mean trees that are as tall as buildings? Or that are taller than buildings? Can trees that are taller than buildings be implemented in this model?

>> It is a drafting error that we corrected: *". . . by trees that are **as** tall than buildings"*
Trees cannot be higher than buildings, this would require modifying radiative calculations. This point is already discussed in Redon et al (2017): *"In the current version of TEB (official SURFEX v8.0), urban trees are assumed to be less tall than surrounding buildings and systematically confined inside the canyon so that they cannot provide shade for roofs. This hypothesis is in accordance with common urban planning specifications for street tree management in Europe (in French, Municipality of Toulouse, 2008; City of Westminster, 2009; Barcelona City Council, 2011)."*

- 11.28: Tree temperature = air temperature ?? It seemed earlier that an energy balance is solved for leaves/ trees, which means a surface temperature is calculated?

>> Conforming to the "bigleaf approach", natural covers (tree and bare soil for TREE-BARE and tree and grass for TREE-GRASS) are combined in a composite natural compartment. Then, the model resolves for this compartment a single SEB, as well as a single temperature evolution equation. This temperature is an average or composite temperature of the natural covers, **but not the temperature of leaves**.
Radiative calculations require to know the leaves temperature. We consequently do the assumption that leaves are in equilibrium with ambient air and their temperature is equal to the air temperature.

- 12.13: Some justification into the selection of the mixing length would be useful. 10 cm as a mixing length seems somewhat low. Maybe scale the mixing length as LAD?

>> This issue related to of the formulation of the mixing length is interesting. However, since we do not have measurements adapted to a specific evaluation, we felt that this section did not provide useful information. We propose to delete this section, since for the time being the formulation of the mixture length remains unchanged compared to that presented by Lemonsu et al (2012).

- Eq. 25: What about the presence of buildings and their effects on mixing length? Is this already included here?

>> Indeed, the height of the buildings and the average frontal are density are used in the calculation of the mixing length (see previous comment about vertical transport)

- Fig.5: Would perhaps benefit from increased vertical resolution.

>> This issue is no more discuss in the manuscript. But we do not think increasing the vertical resolution could really improve the modelling considering the model hypotheses.

- Section 6.4: Which version, with or without the length scale modification, do the authors think is best and consider to be "TEB-Tree"?

>> The formulation of the mixture length remains unchanged compared to that presented by Lemonsu et al (2012).

- Fig. 6: Surely TEB-Tree, by virtue of accounting for shade from trees, should strongly reduce UTCI during daytime in the sun relative to TEB-Ref, which does not account for tree shade?

>> See previous comment about UTCI (General response to all (3) and response to your general comments)

- 13.22: "at night radiation conditions are the same." What about longwave radiation? Surely that differs when near a wall at night? Do you mean to say that at night there is no shortwave radiation?

>> If this is the case, it is because conceptually the calculation of the UTCI for the "sun" case and for the "shade" case is the same except for taking into account the contribution of direct solar radiation (as explained in Section 5). For other sources of radiation received by the individual, there is no difference. Nevertheless, as this part was difficult to understand, we reworked the presentation of the results and the explanation (see general comment/response to all).

- 13.30: "It remains 5 C (warmer?) than in TEB-Ref simulations"

>> The section was reworked for clarification (see general comment/response to all).

- Section 6.5: Regarding the degradation of UTCI at night due to trees – are there measurements of this effect (or of increased longwave from urban tree cover) that you can reference, for example, to indicate that your results are in line with observations?

>> No UTCI measurements were available at the experimental site for comparison. Nevertheless, we completed and improved this section of the article (see response to all) and put our results on UTCI in perspective with those published by Shashua-Bar et al (2011) who calculated indexes of thermal stress from insitu measurements.

---

## Author Response (AR3)

**An urban trees parameterization for modelling microclimatic variables and thermal comfort conditions at street level with the Town Energy Balance model (TEB-SURFEX v8.0)**
by Emilie Redon et al.

**Response to editor's comments**

There seems to be some challenges with the language still (e.g. P1, L1, Model name written wrong). Already Reviewer 1 asked for editing the grammar and language so you should carefully revise the text.
>> We carefully reviewed the entire text. We corrected the typos and as much as possible grammatical and language errors.

Introduction: Reviewer 2 also asked more detailed separation between resolved vegetation and urban canopy models. Text in the intro should be evaluated accordingly. Before UCM there is paragraph where generally the effect of trees is examined but resolved vegetation as such is not discusses. This would fit for example in the start of P2, L29.
>> We have rewritten and considerably completed the introduction. In the first part, we detail the high-resolution models or softwares. Then we present in a second paragraph the urban canopy models for mesoscale applications (which are the models we are interested in). We also focused on the physical processes that can be taken into account by these models and the differences between them.

P7, L13 (or P5, L30): Could you add reference to Lemonsu et al. (2012) for the turbulent fluxes of QH and QE to answer Reviewer1 .
>> The reference to Lemonsu et al. (2012) has been added in Section 3.2 before Eq. 6 and 7.

Reviewer2 was asking how well the current model compares with other models . I see that some text on this should be provided in the Discussion as asked by the reviewer. This text should not empahasize results but rather advantages/differences/disadvatages to the current model compared to others used in the field.
>> Nous avons ajouté un paragraphe de discussion à la fin de la section 6.3 (sur l'évaluation) pour mettre notre approche en perspective avec celles des autres modèles. Néanmoins, toute comparaison reste difficile car les évaluations présentées dans les études précédentes sont souvent limitées en raison du manque de données expérimentales réellement adaptées à l'évaluation détaillée des processus.

Code and data availability: The code used in the calculations of the manuscript should be provided (for re-productivity of the results). Would this be possible to add e.g. as Supplementray material. In general urls should not be used to refer any code and data as they may change in time. The link to SURFEX website is not eg now working. Also the model input and output should be made available either as Supplementary material or under doi created e.g. with Zenodo. And finally also any postprocessing codes used to plot/make data analysis should be included in the same place where input and outputs are made available.
>> We provide the code i.e. the official version of SURFEX v8.0 and the user library corresponding to the new developed code.
> *"Code availability. The new developed code will be included in the next official version of SURFEX v9.0. It is currently a specific user library developed on the basis of the official version SURFEX v8.0. The whole code is available as supplementary material."*

>> The model outputs and R programs to plot them are available as supplementary material. But we do not own the experimental data used for the evaluation. We therefore do not make them available.

[revised manuscript text omitted]

---

## Author Response (AR4)

**An urban trees parameterization for modelling microclimatic variables and thermal comfort conditions at street level with the Town Energy Balance model (TEB-SURFEX v8.0)**
by Emilie Redon et al.

**Response to editor's comments**

Thank you for the modification and corrections. The only thing I'm still missing is the response and action to "Reviewer2 was asking how well the current model compares with other models. **I see that some text on this should be provided in the Discussion as asked by the reviewer. This text should not empahasize results but rather advantages/differences/disadvatages to the current model compared to others used in the field**." in English.

**(sorry we realized our previous reply was in French !!!)**

On this point, we believe we have already provided enough elements in the text. It is said that:
  (1) on the one hand, in terms of physical processes, the model does not separate ground/tree energy balances as it may be the case in other models ;
  (2) on the other hand, in terms of performance (or comparison to observations), it is quite difficult to assess if the other models correctly represent the impact of energy fluxes on air temperature, because the results do not show very substantial differences between sites with and without trees.

It is difficult to go further in the discussion, unless we make an intercomparison of models, and that is not the purpose of this paper (although it would be very interesting given the increase in new parameterizations related to urban vegetation!). We hope you will agree with this answer and this vision.

*"The choice was made to keep the simple approach of "bigleaf" parameterization for treating all natural covers as a composite compartment, rather than solve distinct energy budgets for soil, ground vegetation, and trees. This choice may be debatable, given that there are other models available today that can independently calculate water and energy flows between foliage and air. Nonetheless, in view of the results presented in the literature, it is difficult to objectively compare the performances of these models The main limitation remains the availability of experimental data allowing a fine evaluation of the implemented processes, especially for turbulent flux calculation and impacts on air temperature and humidity (Lee and Park, 2008; Lee, 2011; Ryu et al., 2016). Besides, the comparisons presented by Ryu et al. (2016) do not show a very substantial change in the fluxes and Bowen ratio with the implementation of a specific trees' parameterization."*